# Electrochemical and Structural Characterization of Lanthanum-Doped Hydroxyapatite: A Promising Material for Sensing Applications

**DOI:** 10.3390/ma16134522

**Published:** 2023-06-22

**Authors:** Rocco Cancelliere, Giuseppina Rea, Laura Micheli, Pietro Mantegazza, Elvira Maria Bauer, Asmaa El Khouri, Emanuela Tempesta, Angela Altomare, Davide Capelli, Francesco Capitelli

**Affiliations:** 1Dipartimento di Scienze e Tecnologie Chimiche, Università degli Studi di Roma Tor Vergata, Via della Ricerca Scientifica, 00133 Rome, Italy; laura.micheli@uniroma2.it (L.M.); mantegazza.pietro@gmail.com (P.M.); 2Institute of Crystallography (IC), National Research Council (CNR), Via Salaria Km 29.300, 00016 Rome, Italy; giuseppina.rea@ic.cnr.it (G.R.); davide.capelli@ic.cnr.it (D.C.); 3Institute of Structure of Matter (ISM), National Research Council (CNR), Via Salaria Km 29.300, 00016 Rome, Italy; elvira.bauer@ism.cnr.it; 4Faculté des Sciences Semlalia, BP 2390, Université Cadi Ayyad, Marrakech 40000, Morocco; elkhouriasma@gmail.com; 5Institute of Environmental Geology and Geoengineering (IGAG), National Research Council (CNR), Via Salaria Km 29.300, 00016 Rome, Italy; emanuela.tempesta@igag.cnr.it; 6Institute of Crystallography (IC), National Research Council (CNR), Via Amendola 122/o, 70100 Bari, Italy; angela.altomare@ic.cnr.it

**Keywords:** hydroxyapatite, lanthanum, screen-printed electrodes, electron transfer process, rare earths-based sensors, characterization

## Abstract

In the quest to find powerful modifiers of screen-printed electrodes for sensing applications, a set of rare earth-doped Ca_10−x_RE_x_(PO_4_)_6_(OH)_2_ (RE = La, Nd, Sm, Eu, Dy, and Tm and x = 0.01, 0.02, 0.10, and 0.20) hydroxyapatite (HAp) samples were subjected to an in-depth electrochemical characterization using electrochemical impedance spectroscopy and cyclic and square wave voltammetry. Among all of these, the inorganic phosphates doped with lanthanum proved to be the most reliable, revealing robust analytical performances in terms of sensitivity, repeatability, reproducibility, and reusability, hence paving the way for their exploitation in sensing applications. Structural data on La-doped HAp samples were also provided by using different techniques, including optical microscopy, X-ray diffraction, Rietveld refinement from X-ray data, Fourier transform infrared, and Raman vibrational spectroscopies, to complement the electrochemical characterization.

## 1. Introduction

Hydroxyapatite Ca_10_(PO_4_)_6_(OH)_2_ (HAp) is a phosphate mineral largely retrieved as geomaterial from Earth rocks [1], and it is also the principal inorganic component of the extracellular matrix of bone tissue and of teeth (dentin and enamel) [2]. The wide distribution in biological systems and intrinsic features of HAp (acid-base features, ion-exchange capability, biocompatibility, and adsorption capacity) attracted the interest of materials scientists promoting the research and development of synthetic analogues with customizable properties. The possibility of introducing cationic and anionic substitutions within the HAp framework [3] enables the production of materials with novel characteristics (increased biocompatibility, bioactivity, osteoconductivity, and reduced toxicity and inflammatory nature compared to natural Hap), paving the way to applications in different research fields including, among others, biomedicine, regenerative medicine, and imaging [4,5,6,7,8]. As an example, our previous studies demonstrated the capability of Sr-substituted HAp nanoparticles to regulate molecular and metabolic processes, and to control the spatial and temporal distribution of novel HAp deposits during differentiation of mature stem cells, strengthening a useful function as new drugs for bone healing [9,10,11].

Recently, HAp has also emerged as a useful functional material in the fabrication of sensors/biosensors due to its high biocompatibility and good adsorption properties, which shape a suitable microenvironment for the immobilization of biomolecules, e.g., enzymes over an electrochemical sensor surface [12,13,14]. Vladislavić and co-workers demonstrated that the modification of glass carbon electrodes using in situ synthesized HAp determined improved sensitivity and selectivity towards cysteine oxidation (a non-essential amino acid occurring in protein food) thanks to the excellent absorption properties of the HAp nanoporous structure [15]. Similarly, in gas sensors’ construction, a composite material made of HAp and nano-TiO_2_ provided superior analytical performance compared to nano-TiO_2_ alone in detecting low levels of methanol, ethanol, and propanol vapors [16]. Another valuable possibility is the exploitation of substituted HAp, for example the use of Na-HAp, where the sodium is introduced to decrease the resistivity of the stoichiometric HAp [17].

HAp was also successfully used to determine tyrosine, uric acid, folic acid, L-dopa, and Hg using electrochemical sensors and biosensors [18,19,20,21]. More recently, the same research group tested cerium-substituted HAp with a view to developing new electrochemical sensors to determine norepinephrine, uric acid, and tyrosine from spiked human blood serum and urine samples, with improved analytical parameters [22].

Among the possible doping elements, rare earth (RE) elements received attention as promising enhancers of bioelectrocatalytic activity of enzymes for biosensors fabrication due to their physicochemical properties [23].

This indication prompted the present effort to identify the most effective RE-doping elements for the development of future electrochemical biosensors.

In this work, we perform an electrochemical characterization of a set of RE^3+^—doped Ca_10−x_RE_x_(PO_4_)_6_(OH)_2_ hydroxyapatite powders (RE = La, Nd, Sm, Eu, Dy, and Tm and x = 0.01, 0.02, 0.10, and 0.20) previously obtained in our laboratory using solid-state synthesis, via electrochemical impedance spectroscopy (EIS) and cyclic and square wave voltammetry (CV and SWV) techniques. To date, there is no information on this type of characterization of RE-HAp samples immobilized on screen-printed electrodes. Furthermore, we performed a comprehensive structural characterization of the set of HAp samples doped with the most promising RE, i.e., lanthanum, using optical microscopy, X-ray diffraction from powder data (PXRD), Fourier transform infrared (FTIR), and Raman spectroscopies. This investigation is framed alongside our works on structural characterization and possible applications for calcium phosphate materials.

## 2. Materials and Methods

### 2.1. Synthesis

The Ca_10−x_RE_x_(PO_4_)_6_(OH)_2_ hydroxyapatite samples investigated in the present study are labelled as: HAp = undoped hydroxyapatite and RE1-HAp to RE4-HAp = rare earth-doped hydroxyapatite, with x_RE_ = 0.01, 0.02, 0.10, and 0.20.

The RE-HAp samples were prepared through solid-state synthesis: stoichiometric amounts of CaCO_3_, CaHPO_4_, and RE_2_O_3_ (RE = La, Nd, Sm, Eu, Dy, and Tm) were manually ground and merged in an agate mortar for 1 h; 1 g of each sample of the resulting homogenized powders was then placed in alumina crucibles and calcinated at high temperature (T = 1300 °C) for 7 h [5,24].

### 2.2. Screen-Printed Electrodes and Electrochemical Apparatus

Screen-printed electrodes (SPEs) were in-house manufactured with a 245 DEK screen-printing machine (High performance multi-purpose precision screen printer, Weymouth, UK). These devices are three electrode systems containing a silver-based reference electrode, and graphite-based counter and working electrodes (WE). The electrochemical analyses CV, SWV, and chronoamperometry, were carried out using a PalmSens 4 (Palm Instrument, Houten, The Netherlands).

### 2.3. Preparation of HAp Suspensions and HAp-Modified SPEs 

1,4-dioxane (DO), distilled water (H_2_O), ethanol (EtOH), and mixtures of the last two (H_2_O: EtOH 1:1. 2:1. 3:1. 4:1) were studied as possible solvents for preparation of HAp suspensions. The suspensions (1 mg mL^−1^) of HAp were prepared using the Hielscher UP200St-Ultrasonic Transducer (200 W, 26 kHz, 30 min) (Hielscher Ultrasonics, Teltow, Germany) and then used for functionalization of the (WE). 

Before modification, SPEs were pre-treated using chronoamperometry (1.7 V, 180 s) using a 0.05 M phosphate buffer + 0.1 M KCl pH 7 and then rinsed with deionized water to remove salt residues. WE was modified by drop-casting using 6 µL of 1 mg mL**^−1^** HAp and substituted HAp suspensions.

### 2.4. Analytical Calculations

The electrochemical properties of HAp deposited on SPEs were investigated using EIS and voltammetry. Regarding the EIS measurements, the ΔRct values were determined as follows [25]:(1)$$\Delta \mathrm{Rct}={\mathrm{Rct}}_{\mathrm{blank}}-{\mathrm{Rct}}_{\mathrm{HAp}}$$
where Rct_blank_ (blank HAp) and Rct_HAp_ (doped-HAp) are the charge transfer resistance measured on electrodes before (bare) and after modification with RE-doped Hap, respectively. 

The percentage increase in the signal (faradic current or charge transfer resistance) estimation was calculated using the equation below [26]:(2)$$S\%=\left(\frac{S_{\mathrm{modified}}-S_{\mathrm{bare}}}{S_{\mathrm{bare}}}\right)\times 100$$
in which S_modified_ and S_bare_ correspond to the signal, current peak, or charge transfer resistance value obtained using HAp-modified SPEs and bare SPEs.

### 2.5. Scanning Electron Microscopy

Scanning electron microscopy (SEM) micrographs of La-doped HAp samples were achieved using a Zeiss EVO MA 10 (Jena, Germany), with acceleration voltage = 20 kV and work–distance = 11.5 mm, graphite sample-holder. 

### 2.6. Optical Microscopy/Raman spectroscopy

Morphological and Raman investigations on HAp samples were achieved by using a Malvern Morphology GS3Id (Malvern Panalytical, Malvern, UK), an optical microscope fitted with a 500 mW Raman spectrometer (Kaiser Optics at 785 nm, Kaiser Optical Systems, Inc., Ann Arbor, MI, USA) with a 2 μm spot. The Raman analysis was achieved with acquisition times of 240 s, in the range 100–1800 cm^−1^, and resolution of 4 cm^−1^.

The optical microscope is suitable for the qualitative and quantitative analysis of particle size and particle shape in a range from 0.5 μm to several millimeters. In addition, an analyzer and polarizators can be manually integrated within the optical circuit. Scan mode is executed by using piezoelectric motors which assure the x–y–z movements with accuracy and repeatability of 1 μm. Particle size distribution analysis was achieved using the dedicated software Morphology G3 (User manual, issue 5, version MAN0410). 

### 2.7. Infrared Spectroscopy

FTIR spectra were registered in the spectral range of 400–4000 cm^−1^ (resolution 4 cm^−1^) using a Shimadzu Prestige-21 FTIR instrument (Shimadzu Scientific Instruments, 7102 Riverwood Drive, Columbia, MD, USA) supplied with an attenuated total reflectance (ATR) diamond crystal accessory (Specac Golden Gate).

### 2.8. Powder X-ray Diffraction

The Rigaku RINT2500 diffractometer, having a silicon strip Rigaku D/teX Ultra detector, was used to record PXRD data under the following experimental conditions: room temperature, 50 K, 200 mA, Debye–Scherrer geometry, monochromatic Cu Kα1 radiation (λ = 1.54056 Å) selected using an asymmetric Johansson Ge (111) crystal, step size of 0.02° (2θ), counting time of 4 s/step, 2θ angular range of 8–120° (La1-HAp; La2-HAp; and La3-HAp), and 6–120° (La4-HAp) (2θ), transmission mode. Table 1 reports the main acquisition parameters. To reduce the possible effects of the preferred orientation of crystallites, a special glass capillary with a 0.5 mm internal diameter was filled with the sample and put in rotation on the axis of the goniometer. The software EXPO2013 [27] was run to carry out all the steps of the structure solution process working in the reciprocal space: the determination of the unit cell parameters and the space group, the solution using direct methods, and the Rietveld refinement. Results were obtained using default runs of EXPO 2013. In particular, the N-TREOR09 software [28] embedded in EXPO2013 provided cell parameters. PXRD data were also qualitatively analyzed through QUALX2.0 software [29] based on the PDF-2 database [30] and the free POW_COD database [29]. Further crystal structure information (atomic positions, bonds, angles, etc.) may be retrieved from the joint CCDC/FIZ Karlsruhe online, by quoting the deposit number CSD2250357 (La1-HAp), CSD2250358 (La2-HAp), CSD2250362 (La3-HAp), and CSD2250363 (La4-HAp).

## 3. Results and Discussion

Electrochemical sensors are analytical devices enabling the detection and quantification of a target analyte in complex matrices and have found applications in a variety of disciplines, including medicine, agri-food, and environmental science. To achieve high sensitivity and selectivity, which are required for high-performance sensors, it is frequently necessary to modify electrodes with ad hoc materials. In this study, a series of experiments aimed at optimizing the working protocols and identifying the most effective RE elements for obtaining reliable doped HAp-based electrochemical platforms were conducted.

### 3.1. Electrochemical Characterization

#### 3.1.1. Electrochemical Characterization of HAp—Modified SPEs

To determine the most suitable solvent for enhancing the electroactivity properties of HAp dispersions, various solutions were prepared, deposited on SPEs using the drop-casting technique, and then analyzed [31]. The characterization was carried out by using CV as an analytical tool and [Fe(CN)_6_]^3−/4−^ as a redox couple. The histogram reported in Figure 1 demonstrates that a solution of EtOH:H_2_O (1:2) produced the most effective suspension because the maximum pick current and lowest RSD% (6%) were observed in this condition.

The electrochemical characteristics of differently doped RE-HAp samples were then evaluated by examining five distinct SPEs for each RE-HAp sample. The results of La, Nd, Sm, Eu, Dy, and Tm-modified SPEs are summarized in Table 2.

Among all RE-doped HAp tested, those doped with La, Nd, Sm, and Tm exhibited the best enhancing behavior once deposited on screen-printed platforms, as indicated by the analytical parameters reported in Table 2. Indeed, a significant percentage increase in the signal (always greater than 60%) was observed in terms of both faradic current and ΔRct. However, despite showing the greatest improvement, Nd and Tm have very low reproducibility (RSD% > 20%), which is unacceptable for screen-printed based sensors (RSD limit is 15%). These HAp-related experiments were repeated with identical outcomes (10 electrodes for each type of Nd and Tm-based HAp). Consequently, no additional analyses were conducted on them. Instead, La2-HAp and La4-HAp, and Sm1-HAp and Sm3-HAp displayed electron transfer improvement associated with excellent reproducibility (RSD% < 10). For La2-Hap, La4-Hap, Sm1-Hap, and Sm3-Hap, a net I% of 79, 145, 111, and 131% was observed along with conductivity improvements (ΔRct < 0) of 85, 121, 102, and 101%, respectively. This behavior is also supported by the smaller peak-to-peak separation and peak ratio (Ipa/Ic = 1) between the anodic and cathodic peaks. As expected, the redox probe’s behavior in this condition resembles that of an ideal-reversible couple. Moreover, the voltammograms and the Nyquist plots for the better performing HAp—modified platforms are reported in Figure 2. 

No notable results were obtained for Dy and Lu-based HAp samples as demonstrated from the negative effect occurring at the screen-printed electrodes once modified with their dispersions (%I and %Rct < 0).

#### 3.1.2. Stability and Reusability of La-HAp Modified Platforms

Once the electron transfer properties of all RE-doped HAp samples had been studied, a preliminary study of repeatability, reproducibility, stability, and reusability of the La-HAp modified platforms was performed (relative voltammograms reported in Appendix A).

The study of the stability (reported in Appendix A) was realized by storing La-HAp-modified-SPEs at room temperature in a dark box for 20 days and then measuring the current output using ferro-ferricyanide as electroactive probe and CV as an analytical tool. An almost constant response for up to fifteen days after their fabrication (similar results in terms of registered faradic current) was recorded. In addition, the reproducibility of these platforms was investigated. Precisely eight electrodes made using a uniform procedure were tested using CV analyzing 10 mM [Fe(CN)_6_]^4−/3−^ as an electroactive probe. A good reproducibility with an RSD% of 6% was obtained. In addition, the reusability and sensitivity of platforms modified with La-HAp were studied. For the first one (results presented in Appendix A), encouraging reusability was observed. In fact, an RSD% of 12% was produced by utilizing a 9-step measuring and washing process on the identical electrode. Concerning the sensitivity, a preliminary examination, depicted in Appendix A, was conducted by voltammetrically analyzing ferro-ferricyanide concentrations ranging from 0 to 5 mM (see Appendix A). La-HAp modified platforms showed to be 10 times more sensitive than bare electrodes, thus, demonstrating the good analytical robustness of this platform.

We also performed a comprehensive morphological and structural characterization of the La-doped HAp as reported in the next sections. This characterization was motivated by the lack of exhaustive PXRD structural solutions on La—and in general RE-substituted HAp, apart from some previous works conducted by us on Eu-doped HAp [5,32]. For a complete overview of rare earth distribution within hydroxyapatite structural sites, we quote a work on structure solutions from single-crystal XRD data achieved by Fleet and coworkers [33] on La-, Nd-, Sm-, and Dy-doped HAp.

### 3.2. Morphological Study

High-resolution SEM micrographs run on La-doped HAp samples show the presence of grains with non-regular subspherical shape and size ranging from 5 to 10 μm (Figure 3a–c). Grains with sizes exceeding 20 μm are less common, but they can be observed in Figure 3a (La1-Hap) and Figure 3b (La2-HAp).

Analyses of particle diameters were conducted using scanning optical microscopy (as described in the Materials and Methods section), which is based on the scanning of a large number of particles randomly scattered on an optical glass slide. All particles are first photographed, normalized to a sphere (‘circle equivalent’—CE), and finally analyzed with the software Morphology G3. The CE distribution plots of La-HAp powders are reported in Figure 4a. Regular gaussian distributions are observed showing values in the range of 3–6 μm as the most frequent across the five samples. On closer inspection, we can see that the CE values increase as a function of La % within the HAp framework, from the undoped sample, which shows its maximum at 3 μm, to the La4-HAp samples, which show a peak at 6 μm. All distributions decrease up to 40–50 μm, after which they go to zero value. Figure 4b reports the surface area (A), expressed in μm^2^. Similarly to CE distribution, A values increase as a function of La % within the HAp framework: this is experimentally quite crucial for potential electrochemical applications of such materials because high surface area can influence the electrocatalytic activity of rare earth-doped HAp [22]. Figure 4c,d depict, respectively, aspect ratio, namely the width/length ratio, and the high sensitivity (HS) circularity, defined as = 4 μA/P^2^ (A = area and P = perimeter). Both these dimensionless parameters aim to quantify how close the particles are to a perfect circle, showing the higher values in the two statistics for pure HAp phase at 0.8% of aspect ratio (Figure 4c) and 0.9% of circularity (Figure 4d). La-doped samples seem to have less regular circular shapes, with very similar distribution except for La3-HAp circularity, which is quite similar to the analogous distribution of pure HAp (Figure 4d). Further information is reported in the Appendix A, e.g., optical microscopy photos (Appendix A) and qualitative energy dispersive spectroscopy (EDS) results (Appendix A).

### 3.3. Structural Characterization

#### 3.3.1. PXRD Qualitative Investigation

The typical hexagonal unit cell of HAp [5,8] was identified for our HAp samples. Qualitative analysis revealed the presence of some diffraction peaks, attributed to Ca_3_(PO_4_)_3_ tricalcium phosphate (TCP) and CaCO_3_ calcium carbonate phases. TCP occurrence in HAp synthesis is known from the literature [34], which reports a partial transformation of HAp into TCP at high temperature (900°). The presence of CaCO_3_, observed in the sample La3-HAp, can be interpreted as some unreacted starting materials. The crystallinity of all the samples, calculated according to [35] was, respectively, 99.14, 99.02, 98.78, and 99.07% (from La1-HAp to La4-HAp), while the crystal size was 122.44, 102.90, 100.47, and 124.41 µm. Such high values of crystallinity are, moreover, observed in other HAp products coming from solid-state synthesis [5,34,36]. Figure 5 reports the PXRD experimental pattern of the La3-HAp sample. 

In Figure 6, the experimental diffraction pattern of the La3-HAp sample is shown background-corrected up to 65° 2θ, with the presence of indexed TCP and CaCO_3_ peaks. Other La-HAp samples hold similar PXRD profiles, with no calcium carbonate presence and with TCP occurrence.

The hexagonal *P*6_3_/*m* space group was identified for all the HAp samples [40,41] from the analysis of PXRD data. HAp crystallizes rarely also in monoclinic space group *P*2_1_/*b*; the structural relationships among the two polymorphs were investigated by [42], highlighting a reversible transition monoclinic—hexagonal symmetry at 211 °C: unit cell values increase as a function of the lanthanum doping specie, with cell volumes ranging from 528.36(4) Å^3^ (La1-HAp) to 529.06(4) Å^3^ (La4-HAp) (Table 1). However, considering the standard deviation values, the small increase in the La-doping from La1-HAp to La2-HAp does not evidently change their volumes and unit cell parameters.

The structure model reported in [40] was confirmed for all the HAp samples using the direct methods solution process executed using EXPO. The obtained structural models were refined using the Rietveld method, assuming that the dopant lanthanum species can occupy both calcium sites. Main crystal data are provided in Table 1. A common refinement strategy was adopted: the positions and the displacement atomic parameters of Ca and La sharing the same positions were constrained to be equal and their sum of occupancies was fixed to the value derived from the experimental crystal chemical formula; constraints were applied to the displacement atomic parameters of P and O atoms set to be equal. The Fourier analysis was unable to locate hydrogen atom positions that were placed exploiting the information about the structural model of [40]. Table 1 provides the main crystal structure refinement data. The presence of a high background signal at small 2θ angles in all the La-HAP samples and of an additional crystalline phase, even though with a small percentage, prevented the refinement process from providing high-precision results and lowered the reliability of the structural characterization presented in this paper. Figure 7 displays the agreement between the observed (blue line) and the calculated (red line) diffraction pattern for the La2-HAp sample; the background (green line) and the difference pattern plotted on the same scale (violet line) are shown. Similar plots hold for other La-doped HAp samples.

#### 3.3.2. Structural Arrangement of La-Doped HAp

There is a vast and exhaustive bibliography on structural characterization of undoped hydroxyapatite samples [8,34,40]; for this reason, in the present work, we focus only on structural features of La-doped HAp, of which to date there are no issues in the literature. The discussion is fixed according to the structural arrangement of hexagonal *P*6_3_/*m* HAp, with atoms occupying different crystallographic sites, i.e., Ca1 in 4*f* special site, Ca2, P1, O1, and O2 on 6*h* special site, O3 on 12*i* general position, and O4_OH_ on 4*e* special site (Table 3) [43]. 

All bond distances found in the present La-doped HAp structural models are listed in Table 4, together with the results of the analysis of bond valence parameters [44]. The P(1)O_4_ phosphate group shows the regular tetrahedral coordination (Figure 8) typical of inorganic orthophosphates [45,46,47], with P–O distances in the range 1.50–1.60 Å: such values, coming from powder XRD structure solutions, as also observed in other PXRD works on inorganic phosphates [5,48,49], are somewhat longer than analogous values coming from single-crystal XRD structural solutions [50].

The Ca atoms display two different complex coordinations, i.e., Ca(1)O_9_ polyhedron, with three different pairs of bond distances related by site symmetry, and Ca(1)O_6_(OH), which can be described by an irregular pentagonal bipyramid characterized by five bonds (one Ca2-O1 and two different pairs of symmetry-related Ca2-O3 bonds) on the equatorial plane, and the two vertices occupied by O2 and the hydroxyl group (Figure 8). Ca-O bond distances are usually between 2.40 and 2.85 Å [50]; Ca-O distances over 2.8 Å, typical of Ca1 site, are considered out of the bonding sphere of calcium, displaying weak bonding character, as shown in bond valence parameters analysis (Table 4): for this reason Ca1 polyhedron is also indicated as Ca(1)O_6_ metaprism (polyhedron ideally intermediate between octahedron and trigonal prism), discarding Ca1-O3 distances as bonds and considering them as interactions.

The O4 hydroxyl atoms are placed on the 4*e* site, displaying disordered site distribution, i.e., split above or below the mirror plane (*m*) [43]. The present feature leads to a local deviation from the hexagonal symmetry, with a consequent lack of the *m* plane, given that only one of the two *m*-related sites is statistically occupied. However, with each *m*-related site occupied at 50%, the average hexagonal P6_3_/*m* setting is retained [1]. OH is bound to Ca2 cation, with distances in the range 2.60–2.80 Å (Table 4).

Hexagonal *P*6_3_/*m* hydroxyapatite, whose crystal formula can be indicated as [Ca(1)_4_Ca(2)_6_](PO_4_)_6_(OH)_2_, displays a zeolitic character, due to the presence of cavities (channels) typical of these silicates [51]: in detail, the HAp three-dimensional arrangement resembles a lattice made of arrays of face-sharing Ca1O_6_ metaprisms, corner-connected to PO_4_ tetrahedral groups down the *c* crystallographic axis; the result of this arrangement is the formation of one-dimensional cavities filled by [Ca(2)_6_(OH)_2_]^10+^ moieties. The Ca1O_6_ metaprism is ideally built up by six Ca-O bond distances up to 2.8 Å; after this threshold, the further three symmetry-related Ca1-O3 distances are considered interactions, which contribute to stabilize the framework. The HAp lattice has ideal stoichiometry [Ca(1)_4_(PO_4_)_6_]^10−^, balanced by [Ca(2)_6_(OH)_2_]^10+^ moieties located in the hexagonal cavities. Figure 9 reports the 3-D framework of HAp down the crystallographic axis *c*.

The distribution of rare earths within the HAp structural sites was examined by looking for the possible occurrence of lanthanum in the two calcium structural sites, and also owing to the results of bond valence parameters investigations [44]. A better response in terms of agreement indices indicated the presence of RE in the Ca2 site, as described by [33,52]. Refined occupancy values for calcium and lanthanum for the Ca2 site were Ca = 0.998(3)/La = 0.002(3) for La1-HAp, 0.994(1)/0.006(1) for La2-HAp, 0.988(2)/0.009(3) for La3-HAp, and 0.955(3)/0.045(3) for La4-HAp. The precision of the variables refined using the Rietveld method, in particular of the La occupancy for La1-HAp, is poor due to the small doping percentage, as well as the presence of a high background signal at small 2θ angles and an additional crystalline phase, making questionable the presence of La in the lowest doped sample. Calculated bond valence sum (bvs) values for the Ca2 site exceed the ideal value of 2.00 valence units, especially in the samples with the highest doping ratio (La3-HAp, La2-HAp), suggesting the presence of a cation with oxidation number 3+ partially replacing Ca2. Outcomes of qualitative bond valence investigation are provided in Table 4.

#### 3.3.3. Vibrational Spectroscopy (FTIR, Raman)

The FTIR spectra of La-doped HAp powders were registered in the 400–4000 cm^−1^ wavenumber region, and are depicted in Figure 10, compared with the spectrum of pure hydroxyapatite. Experimental band positions (wavenumbers, cm^−1^) are provided in Table 5.

FTIR vibrational spectroscopy is widely used in phosphate characterization, owing to its accuracy in volatile and light elements detection, and its complementarity with XRD information [53,54,55].

Accordingly, FTIR spectroscopic investigations of HAp samples are reported for a long time in the literature [56,57,58,59], and in recent years works on rare earth-doped HAp [60,61], comprising also lanthanum substitution [62,63,64,65,66] have been published. HAp is characterized by an appreciable molecular nature with typical (PO_4_)^3−^ vibrational modes, even if usual, the description of the spectra starts from the peak at 3572 cm^–1^ attributed to the hydroxyl stretching mode, this being the representative peak of HAp [8,42,56,64]. The intensity and shape of the (OH)^−^ stretching vibration diminishes with increasing La content in the samples examined here. The latter phenomenon has been ascribed to the substitution of Ca^2+^ with La^3+^ and as a consequence the decrease in the number of calcium-bonded (OH)^−^ ions which transform into O^2−^ ions [65]. 

A close inspection of the wavenumber region 1500 cm^−1^–1400 cm^−1^ reveals several low intensity vibrations especially in the HAp and La3-HAp samples; such vibrations can be explained by the presence of (CO_3_)^2-^ anions [57,62], thus confirming the PXRD analysis. 

In the lower wavenumber region, the strongest peaks in the 1088–1018 cm^−1^ range are pertinent to the triply degenerated antisymmetric stretching modes (ν_3_) of the (PO_4_)^3−^ phosphate anion [60], while the peak at 960 cm^−1^ is related to the symmetric stretching of the phosphate. The two strong and sharp peaks at 598 and 567 cm^−1^ are due to the triple degenerate antisymmetric bending mode of the (PO_4_)^3−^ group [61]. Investigation of deuterated specimen [56] indicates that the relatively sharp peak at 627–630 cm^–1^ is pertinent to the O-H liberation. The sharpness of selected bands in the phosphate region, especially 627 cm^−1^, 597 cm^−1^, and 562 cm^−1^, is an experimental tool for detection of good crystallinity [5]. Finally, the peak at 472 cm^−1^ can be assigned to the ν_2_(PO_4_)^3−^ mode [58]. Lastly, in samples with a higher concentration of rare earth dopant, the peak at 505–508 cm^−1^ could be assigned to the RE-O mode, as suggested by Serret for La-doped high-temperature HAp phases [66]. 

As a result of increasing the ratio of the lanthanum replacement in HAp samples, no relevant changes in the infrared spectra are observed (Figure 10). In fact, no or only slight shifts of the relative characteristic absorbances of the phosphate anion have been detected. In an case, modest broadening of the spectral features around 1000 cm^−1^,especially towards the higher wavelengths, has been observed. This could suggest an enhancement in local disorder around the (PO_4_) sites as a result of the partial substitution of Ca2 with La. A similar infrared spectrum has already been reported for lead-substituted HAp [67].

Raman vibrational spectroscopic technique has become quite popular in materials characterization in recent years, allowing rapid and non-destructive analysis, and providing fingerprint spectra: also, in this case, there is a vast and exhaustive bibliography on hydroxyapatite characterization [8,68,69,70,71], but less frequent are Raman characterizations on RE-doped HAp: we quote works on Eu-HAp [32,72], and a time-resolved study on synthetic and natural terms of Dy-, Eu-, Nd-, and Sm-HAp [72].

Figure 11 shows the Raman spectra in the region 300–1200 cm^−1^ collected for the La-HAp and undoped samples (HAp), while Table 6 reports the experimental band positions and the pertinent assignments. No additional signals were registered in the other analyzed regions of the spectra. The spectra show the typical emissions related to the modes of vibration of the phosphate anions [61,62,63]. The most intense band, located at about 960 cm^−1^, is related to the symmetric stretching of the (PO_4_)^3−^ groups, whereas the emissions observed from 1029 to 1078 cm^−1^ are caused by the (PO_4_)^3−^ asymmetric stretching vibrations (Table 6) (Figure 11). In the lower wavenumbers region, instead, the bands at ~429 and 447 cm^−1^ are related to the ν_2_(PO_4_)^3-^ vibrations and those between ~580 and 608 cm^−1^ are due to the (PO_4_)^3−^ bending vibrations (Table 3). By incrementing the La ratio in the HAp, the Raman bands related to the vibration of the (PO_4_)^3−^ groups tend to become slightly broadened but are not noticeably shifted in accordance with the observations made in the FTIR investigation. The latter behavior is a consequence of the enhancement of the structural disorder, as previously reported in [70].

## 4. Conclusions

In this study, an in-depth electrochemical and morphological characterization of RE-doped HAp samples was carried out.

The electrochemical characterization conducted using EIS, CV, and SWV suggested La-HAp as the most promising material for sensing applications. Indeed, an important improvement in terms of sensitivity, repeatability, and reusability was assessed for SPE-modified with La-doped HAp. The morphological characterization of the latter, based on optical microscopy, highlighted an important increment of surface area of HAp particles, therefore, explaining the results of the electrochemical analysis. The structural characterization, based on X-ray diffraction, FTIR, and Raman spectroscopies, showed increasing unit cell dimensions according to PXRD data, and a band mode in FTIR spectra at 505–508 cm^−1^ reasonably assigned to the RE-O bond.

This work represents an important advancement in understanding the effect of RE-based HAp once used for electrochemical purposes. Indeed, their analytical robustness is particularly intriguing, indicating that their use in the development of sensors is a sound concept.

## Figures and Tables

**Figure 1 materials-16-04522-f001:**
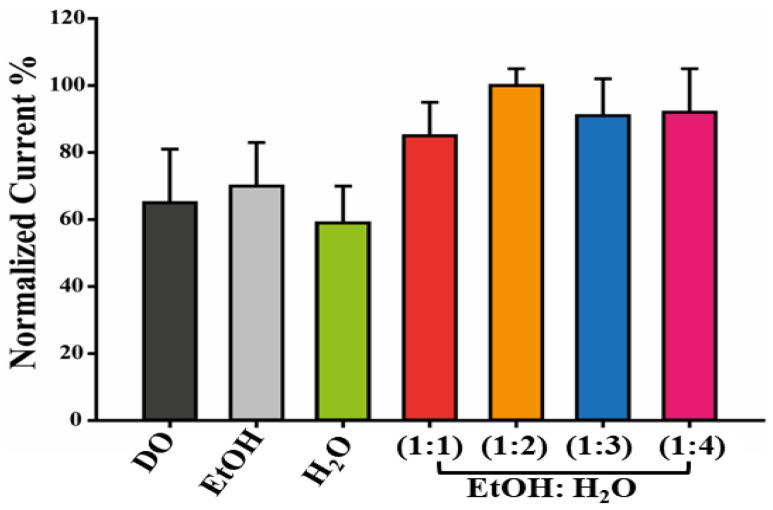
Dispersions optimization. Histogram reporting the current recorded using HAp dispersed in 1,4-Dioxane (DO), Ethanol (EtOH), Water (H_2_O), and EtOH: H_2_O mixture (1:1, 1:2, 1:3, 1:4 vv^−1^) to modify the WE of SPEs.

**Figure 2 materials-16-04522-f002:**
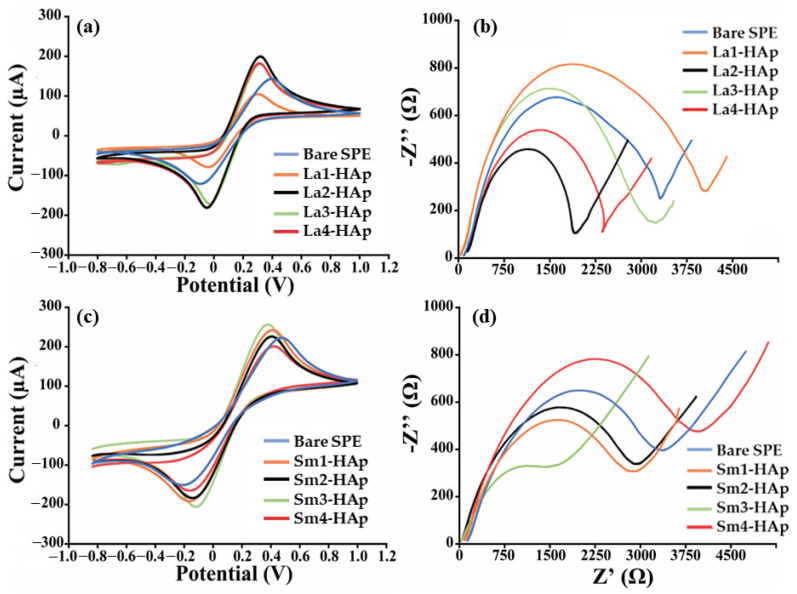
Electrochemical characterization of La- and Sm-doped HAp-modified SPEs. (**a**,**c**) Cyclic voltammograms and (**b**,**d**) Nyquist’s plot recorded in 10 mM [Fe(CN)_6_]^4−/3−^ in 50 mM PBS using La-HAp- and Sm HAp-modified SPEs.

**Figure 3 materials-16-04522-f003:**
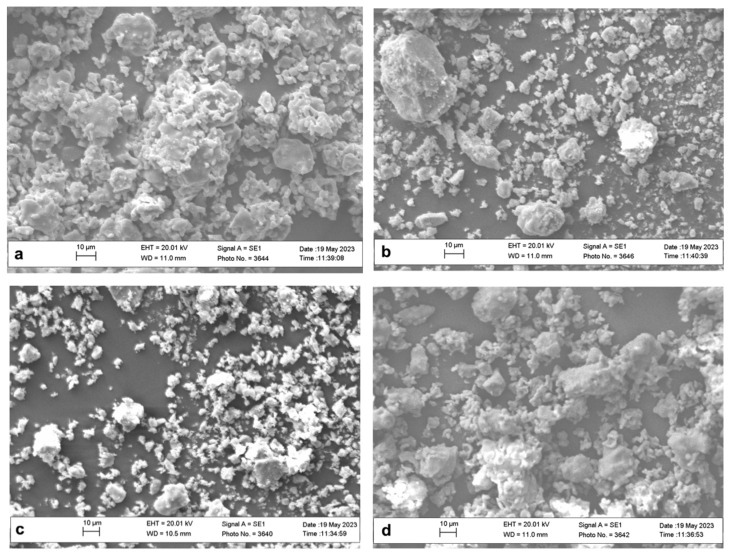
SEM micrographs of La-doped HAp samples: La1-HAp (**a**), La2-HAp (**b**), La3-HAp (**c**), and La4-HAp (**d**).

**Figure 4 materials-16-04522-f004:**
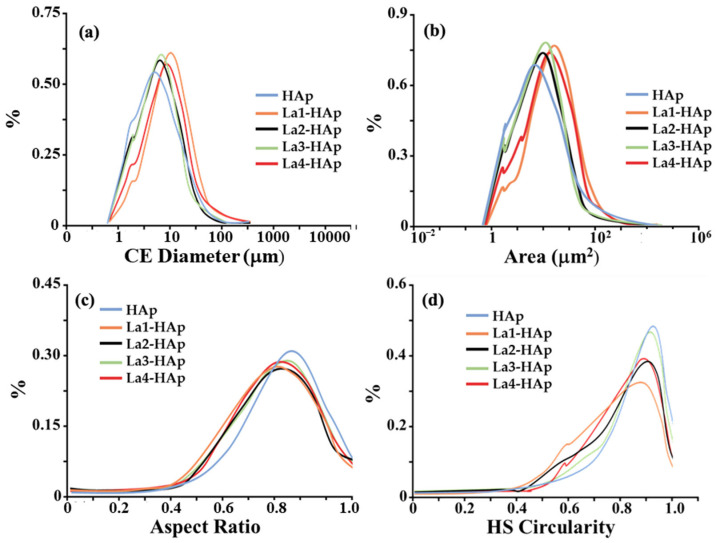
Morphological analyses of pure and La-doped HAp samples: (**a**) ‘circle equivalent’ (CE) diameter distribution curve (logarithmic scale); (**b**) particle area (A) (logarithmic scale); (**c**) aspect ratio = width/length ratio; and (**d**) HS circularity = 4 μA/P^2^ (A = area and P = perimeter).

**Figure 5 materials-16-04522-f005:**
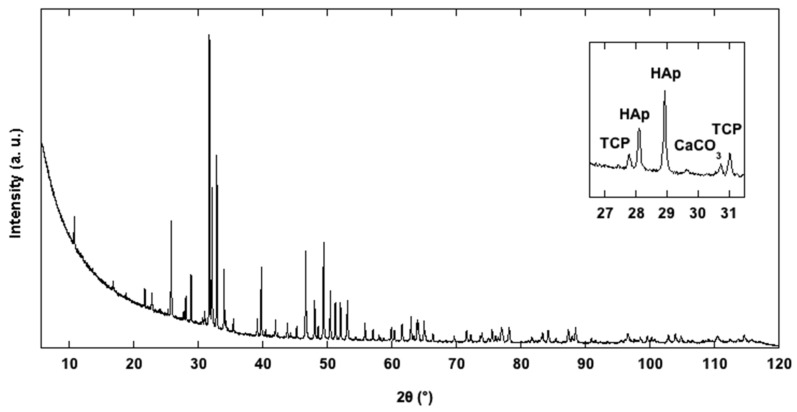
Observed PXRD profile of La3-HAp sample.

**Figure 6 materials-16-04522-f006:**
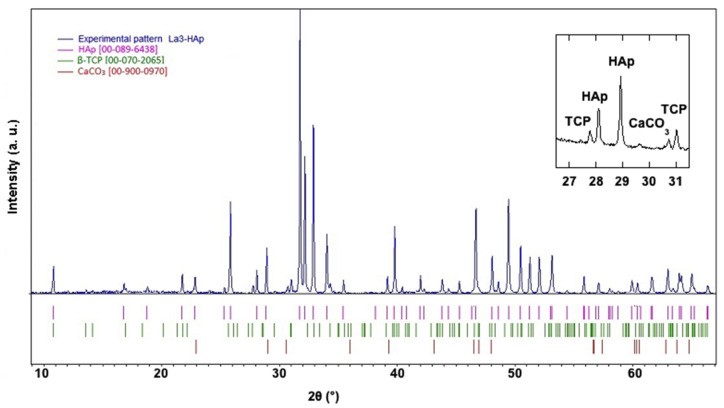
Experimental diffraction pattern background-corrected of La3-HAp sample in the 2θ interval 8–65°, with diffraction lines corresponding to β-TCP (JCPDS, PDF-2, and 00-070-2065) [37,38], CaCO_3_ [COD 00-900-0970] [39], and HAp (JCPDS, PDF-2, and 00-089-6438) [40]. Inset: significant diffraction peaks corresponding in the 2θ interval 26.5–31.5°.

**Figure 7 materials-16-04522-f007:**
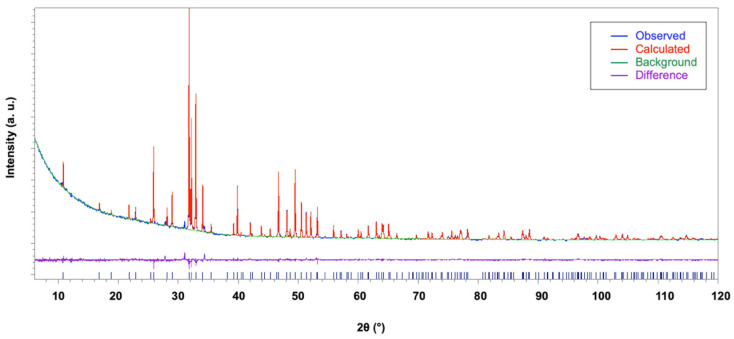
Rietveld plot of the La2-HAp sample. Observed diffraction profile (blue line), calculated profile (red line), background (green line), and difference profile (magenta line). Blue bars: HAp indexed peaks.

**Figure 8 materials-16-04522-f008:**
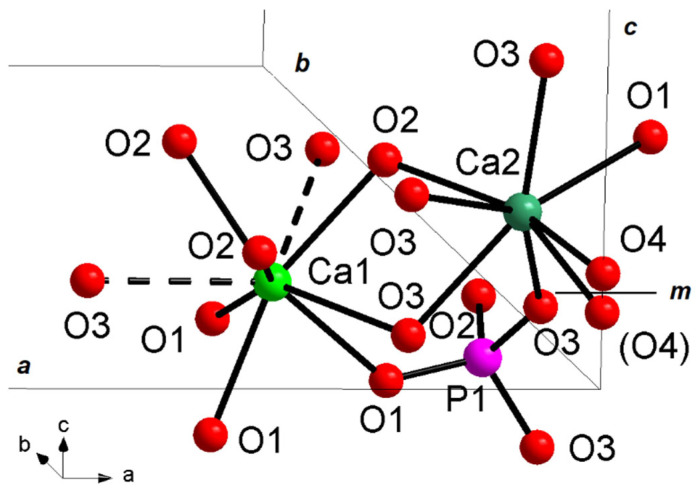
Ca1, Ca2, and P1 structural sites in hydroxyapatite unit cell of La2-HAp sample. Dashed lines: Ca-O interactions > 2.8 Å. *m* = mirror plane. Similar plots hold for other La-HAp samples.

**Figure 9 materials-16-04522-f009:**
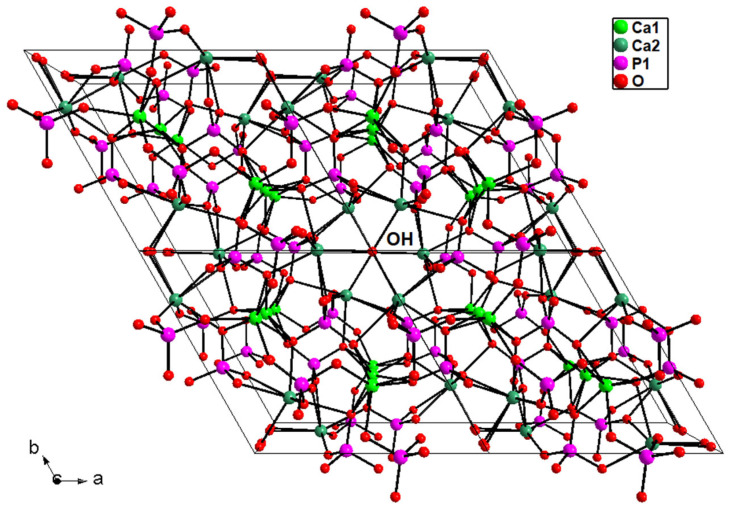
Three-dimensional framework of hydroxyapatite sample La2-HAp viewed down *c*. The hydroxyl groups correspond to O4 atoms. Ca1-O > 2.8 Å bonds not drawn. Similar plots hold for the other La-HAp samples.

**Figure 10 materials-16-04522-f010:**
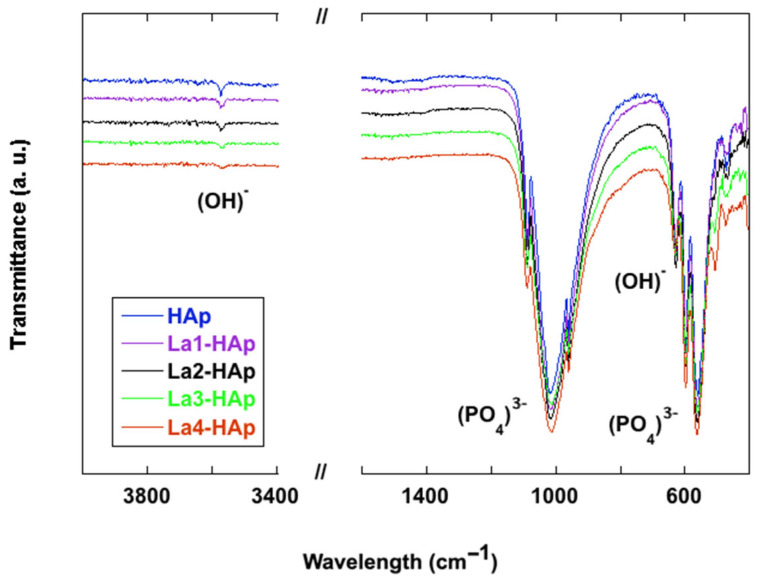
FTIR spectra of La-doped HAp powders compared with pure HAp in the low-frequency (<1200 cm^−1^) lattice mode region. Inset: (OH)^-^ peak at 3572 cm^−1^.

**Figure 11 materials-16-04522-f011:**
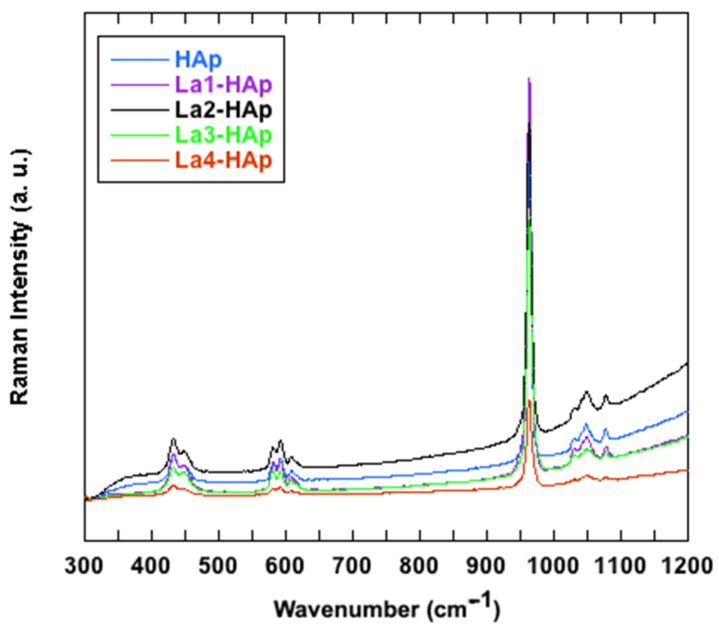
Raman spectra of La- and undoped-HAp, in 300–1200 cm^−1^ range.

**Table 1 materials-16-04522-t001:** Main crystallographic parameters for Ca_10−x_La_x_(PO_4_)_6_(OH)_2_ (x = 0.01, 0.02, 0.10, and 0.20) HAp samples. FMLS: full-matrix least square.

	La1-HAp	La2-HAp	La3-HAp	La4-HAp
Nominal La_x_ Formula wt.	La_0.01_ 1005.61	La_0.02_ 1008.10	La_0.10_ 1011.95	La_0.20_ 1031.30
Color	Colorless	Colorless	Colorless	Colorless
T (K)	293	293	293	293
λ (Å)	1.54056	1.54056	1.54056	1.54056
2θ; step (*°*)	8–120, 0.02	8–120, 0.02	8–120, 0.02	6–120, 0.02
Crystal system Space group	Hexagonal *P*6_3_/*m*	Hexagonal *P*6_3_/*m*	Hexagonal *P*6_3_/*m*	Hexagonal *P*6_3_/*m*
*a* = *b* (Å)	9.4146(3)	9.41472(13)	9.41670(19)	9.4194(3)
*c* (Å)	6.8834(4)	6.88333(18)	6.8835(3)	6.8853(3)
*V* (Å^3^)	528.36(4)	528.377(17)	528.61(2)	529.06(4)
Z; ρ_calc._ (Mg·m^−3^)	1, 3.160	1, 3.168	1, 3.185	1, 3.237
Refinement	FMLS	FMLS	FMLS	FMLS
Bragg refl.	288	288	288	288
R*_p_*; R*_wp_*; R*_exp_* (%)	4.21, 6.81; 6.07	2.21, 3.59, 4.73	2.54, 4.27, 4.91	4.75, 7.63, 5.43

**Table 2 materials-16-04522-t002:** Charge transfer resistance, peak-to-peak separation, anodic and cathodic peak ratio (Ipa/Ipc), and signal percentage increase estimated for Ca_10−x_La_x_(PO_4_)_6_(OH)_2_ (x = 0.01, 0.02, 0.10, and 0.20) HAp samples using CV and EIS in 10 mM [Fe(CN)_6_]^4−/3−^, in 50 mM PBS pH 7.4. Average values of at least 5 SPEs are presented.

Sample	ΔRct (Ω)	ΔE (V)	Ipa/Ipc	% Increase in the Signal	
				Rct	Current
La1-HAp	−640 ± 54	0.55 ± 0.02	1.07 ± 0.1	63	55
La2-HAp	−868 ± 26	0.36 ± 0.08	1.01 ± 0.1	85	79
La3-HAp	−727 ± 91	0.58 ± 0.03	1.07 ± 0.1	71	68
La4-HAp	−1231 ± 65	0.35 ± 0.02	1.01 ± 0.1	121	145
Nd1-HAp	−1583 ± 250	0.32 ± 0.01	1.03 ± 0.1	155	123
Nd2-HAp	−1256 ± 280	0.31 ± 0.02	1.06 ± 0.1	123	56
Nd3-HAp	−1516 ± 320	0.30 ± 0.01	1.02 ± 0.1	149	88
Nd4-HAp	−1127 ± 298	0.30 ± 0.04	1.06 ± 0.1	111	97
Sm1-HAp	−1041 ± 71	0.31 ± 0.02	1.02 ± 0.1	102	111
Sm2-HAp	−605 ±88	0.59 ± 0.03	1.08 ± 0.1	59	43
Sm3-HAp	−1028 ± 51	0.29 ± 0.01	1.01 ±0.1	101	131
Sm4-HAp	−724 ± 100	0.61 ± 0.01	1.05 ± 0.1	71	56
Dy1-HAp	133 ± 21	0.38 ± 0.01	1.00 ± 0.1	−13	−21
Dy2-HAp	183 ±15	0.38 ± 0.02	1.00 ± 0.1	−18	−26
Dy3-HAp	269 ± 23	0.39 ± 0.01	1.01 ± 0.1	−26	−36
Dy4-HAp	164 ± 16	0.39 ± 0.01	0.99 ± 0.1	−16	−13
Tm1-HAp	−970 ± 243	0.56 ± 0.04	1.04 ± 0.1	95	62
Tm2-HAp	−1119 ± 321	0.65 ± 0.001	1.05 ± 0.1	110	76
Tm3-HAp	−1245 ± 211	0.63 ± 0.04	1.07 ± 0.1	122	112
Tm4-HAp	−868 ± 188	0.65 ± 0.01	1.05 ± 0.1	85	93
Lu1-HAp	138 ± 21	0.36 ± 0.01	1.00 ± 0.1	−14	−34
Lu2-HAp	291 ± 19	0.37 ± 0.01	1.01 ± 0.1	−29	−33
Lu3-HAp	289 ± 23	0.36 ± 0.01	1.02 ± 0.1	−28	−19
Lu4-HAp	357 ± 17	0.36 ± 0.01	1.03 ± 0.1	−35	−41

**Table 3 materials-16-04522-t003:** Structural sites in hydroxyapatite.

Atom	Site	x	y	z
Ca1	4*f*	0.6667	0.3333	z
Ca2	6*h*	x	y	0.2500
P1	6*h*	x	y	0.2500
O1	6*h*	x	y	0.2500
O2	6*h*	x	y	0.2500
O3	12*i*	x	y	z
O4_OH_	4*e*	0	0	z

**Table 4 materials-16-04522-t004:** Bond lengths (Å) and bond valence parameters (*bvp*, valence units) for La-HAp samples. Bvps for Ca2 calculated according to La occupancies.

Distance	La1-HAp	*bvp*	La2-HAp	*bvp*	La3-HAp	*bvp*	La4-HAp	*bvp*
3xCa1-O1	2.409(11)	*0.30*	2.412(4)	*0.30*	2.406(6)	*0.31*	2.405(7)	*0.31*
3xCa1-O2	2.445(11)	*0.28*	2.451(3)	*0.27*	2.447(6)	*0.27*	2.445(11)	*0.28*
3xCa1-O3	2.810(9)	*0.10*	2.834(3)	*0.10*	2.846(6)	*0.10*	2.832(7)	*0.10*
		* **2.04** *		* **2.01** *		* **2.04** *		* **2.07** *
Ca2-O1	2.683(16)	*0.14*	2.683(5)	*0.14*	2.691(9)	*0.15*	2.675(11)	*0.18*
Ca2-O2	2.354(13)	*0.35*	2.364(4)	*0.34*	2.372(7)	*0.35*	2.364(7)	*0.39*
2xCa2-O3	2.329(8)	*0.38*	2.335(3)	*0.37*	2.338(4)	*0.39*	2.336(7)	*0.42*
2xCa2-O3	2.507(11)	*0.23*	2.495(4)	*0.21*	2.480(6)	*0.26*	2.483(8)	*0.29*
Ca2-O4_OH_	2.379(5)	*0.33*	2.378(2)	*0.33*	2.365(4)	*0.36*	2.360(5)	*0.39*
		* **2.04** *		* **1.97** *		* **2.16** *		* **2.38** *
P1-O1	1.558(18)	*1.17*	1.527(6)	*1.27*	1.538(10)	*1.24*	1.565(14)	*1.15*
P1-O2	1.541(13)	*1.23*	1.547(4)	*1.21*	1.559(7)	*1.17*	1.559(11)	*1.17*
2xP1-O3	1.546(8)	*1.21*	1.543(3)	*1.22*	1.547(5)	*1.21*	1.552(7)	*1.19*
		* **4.82** *		* **4.92** *		* **4.83** *		* **4.70** *

**Table 5 materials-16-04522-t005:** Experimental absorptions in the infrared spectra with pertinent assignments for present samples.

HAp	La1-HAp	La2-HAp	La3-HAp	La4-HAp	Assignment
3572	3572	3572	3572	3572	ν_s_(OH)
1086	1087	1087	1088	1088	ν_3_(PO_4_)^3−^
1047	1047	1047	1047	1048	
1018	1019	1019	1019	1019	
960	960	960	961	961	ν_1_(PO_4_)^3−^
627	628	628	629	630	ν_l_(OH)
597	598	598	598	598	
562	562	562	562	562	ν_4_(PO_4_)^3−^
-	508	508	506	505	RE-O
472	472	474	474	474	ν_2_(PO_4_)^3−^

**Table 6 materials-16-04522-t006:** Raman band wavenumbers (cm^−1^) and pertinent assignments for HAp samples.

HAp	La1-HAp	La2-Hap	La3-HAp	La4-HAp	Assignment
430	432	432	432	432	ν_2_(PO_4_)^3−^
445	448	448	448	448	
580	581	581	580	581	ν_4_(PO_4_)^3−^
589	592	592	592	592	
606	608	609	608	608	
961	963	962	962	962	ν_1_(PO_4_)^3−^
1029	1030	1030	1030	1030	ν_3_(PO_4_)^3−^
1044	1048	1048	1048	1048	
1073	1077	1077	1077	1077	

## Data Availability

Crystal structure information (atomic positions, bonds, angles, etc.) may be retrieved from the joint CCDC/FIZ Karlsruhe online, by quoting the deposit number CSD2250357 (La1-HAp), CSD2250358 (La2-HAp), CSD2250362 (La3-HAp), and CSD2250363 (La4-HAp).

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
