# Peer review of "Electrochemical and Structural Characterization of Lanthanum-Doped Hydroxyapatite: A Promising Material for Sensing Applications"

_materials, 2023, doi:10.3390/ma16134522_

Round 1

Reviewer 1 Report

The submitted manuscript deals with the comprehensive material characterization of doped hydroxyapatite, which is intended for sensor applications. The results of the study provide new data and insights important for further materials research. The entire text is well understood at a high scientific level, and the data are well explained and interpreted. The manuscript contains no major flaws and I agree to publish it as submitted.

Please check the text on lines 101 and 177 carefully.

Author Response

- Please check the text on lines 101 and 177 carefully.

We thank the Reviewer for the observation. The indicated text was revised.

Reviewer 2 Report

This manuscript deals with synthesis and characterization of electrochemical (the Impedance Spectroscopy, Cyclic and Square Wave voltammetry) and structural (the optical microscopy, powder X-ray diffraction, X-ray
Rietveld refinement, Fourier Transform Infrared and Raman vibrational spectroscopies) of lanthanum-doped hydroxyapatite. Rare earth-doped hydroxyapatites are good candidates to applications in different fields,
including biomedicine and regenerative medicine owing to its biocompatibility, bioactivity, osteoconductivity, and reduced toxicity and inflammatory properties. Indeed, this complex and deep study allowed to suggest that lanthanum-doped hydroxyapatite is highly promising material for sensing applications due to its sensitivity, repeatability and reusability.

I have no significant critical arguments against the publication of this work.
Small remarks:

1) a large number of abbreviations makes it very difficult to read the article.
2) It would be good to discuss the accuracy of the data obtained. The least-square errors given for the structural results obtained by the Rietveld method look greatly overestimated.

Author Response

- a large number of abbreviations makes it very difficult to read the article.

We agree with the Reviewer. We removed three acronyms: RE (reference electrode), CE (counter electrodes) and CA (chronoamperometry) as not necessary. In our opinion, other abbreviations serve to keep the text unburdened.

- It would be good to discuss the accuracy of the data obtained. The least-square errors given for the structural results obtained by the Rietveld method look greatly overestimated.

Taking into consideration the Reviewer's suggestion, we discussed the accuracy of the obtained data by adding the following sentence:

'The presence of a high background signal at small 2 theta angles in all the La-HAP samples and of an additional crystalline phase even though with a small percentage prevent the refinement process from providing high-precision results. This does not lower the quality of the structural characterization presented in this paper.'

Reviewer 3 Report

The  manuscript  “Electrochemical and structural characterization of lanthanum-doped hydroxyapatite: a promising material for sensing applications” by  Rocco Cancelliere et al. describes a  set of  several  Re doped Hap. The emphasis is on the  Electrochemical characterization.

I am not sure if the Electrochemical characterization in solution is  applicable or can be directly  transferred to  “printed” layers.  I don’t see how 1.4-dioxane is  environmentally  friendly.

In the  Abstract the  authors mention that the  properties were  measured before the  characterization. Usually  you need to know  what  is obtained/synthesized  e.g.  structural characterization precedes the  properties investigations.

The PXRD FTIR are nice but have you assessed the  amounts of Re  by WDXRF, XPS, EDX, ort other  chemical  method?

The optical  microscopy part   particle dimensions provides a  CE  of  3 nm (line  253? ).

However  the PXRD  suggest the presence of other phases. How do you interpret?

Expo should have generated a CIF file  with the  refinement. It needs to be provided as SI  (probably along with the “extracted” hkl.    

Authors state La-doped but the  0.002(3) -  0.006 occupancies  need are  basically nothing, the first one  is  actual 2 with error  3.

Probably  optical  fluorescence microscopy  or just fluorescence will  be needed to prove the doping not Raman /FTIR  - the  508 band is  very frail ( compared to   1000 and 630 cm-1.

The weak broadening of Raman peaks for  re presence ?  

I am not sure that RE2O3  (oxide) will be structural incorporated in Hap  . In the synthesis  procedure the  agate mortar for 1 hour; manually or by ball milling ?

homogenized powders were placed in alumina crucibles. Were they  pressed/pelleted? What were the amounts  used e.g. 1 g – 5 g 10 g total ?

XRF/XPS or other methods  providing  0.002 accuracy or better  is  a  must. 

Fluorescence is  also suited.

In its  present form  the  manuscript is not suitable  for  publication.

Author Response

- I am not sure if the Electrochemical characterization in the solution is applicable or can be directly transferred to “printed” layers. I don’t see how 1.4-dioxane is environmentally friendly.

We are not sure to have properly understood this comment. All electrochemical measurements were performed at the RE-doped HAp modified electrode/electrolite interface.

Regarding 1,4-dioxane, it was used only as an alternative dissolving solvent for hydroxyapatite as proposed by Goonasekera et al. (2016) in their work (DOI https://doi.org/10.1039/C5TB02255J). In fact, 1,4-dioxane (DO) was only tested cautiously and found to be ineffective, whereas EtOH:H2O was discovered to be the most acceptable solvent.

- In the Abstract the authors mention that the properties were measured before the characterization. Usually you need to know what is obtained/synthesized e.g. structural characterization precedes the properties investigations.

The Reviewer is right, in fact the electrochemical characterization of RE-doped HAp followed structural investigation. It was anticipated in the manuscript for expositive purposes to focalise the attention on La-doped HAp. The text was modified properly in the abstract.

- The PXRD FTIR are nice but have you assessed the amounts of Re by WDXRF, XPS, EDX, or other chemical method?

We thank the Reviewer for the suggestion, we have performed and included now in the manuscript SEM images and qualitative EDS spectra in the Supporting Informations.

- The optical microscopy part particle dimensions provides a CE of 3 nm (line 253?).

Sorry, it was a mistake. It is 3 μm. We modified the text accordingly.

- However the PXRD suggest the presence of other phases. How do you interpret?

Lines 273-276 in the text provide a brief interpretation of the presence of additional phases. Following the Reviewer’s observation, we improved the interpretation in the text:

'TCP occurrence in HAp synthesis is known from literature [Ref 34], that reports a partial transformation of HAp into TCP at high temperature (900°). The presence of CaCO3, observed in the sample La3-HAp, can be interpreted as some unreacted starting materials.'

-Expo should have generated a CIF file  with the  refinement. It needs to be provided as SI  (probably along with the “extracted” hkl.

We thank Reviewer for this suggestion. Even if the CIF deposit numbers were reported at line 156, now the CIF files are provided in the Supporting Informations.  

- Authors state La-doped but the  0.002(3) -  0.006 occupancies  need are  basically nothing, the first one  is  actual 2 with error  3.

We thank the Reviewer for this comment. We revised the sentence and added text:

'Refined occupancy values for calcium and lanthanum for the Ca2 site were Ca = 0.998(3) / La = 0.002(3) for La1-HAp, 0.994(1) / 0.006(1) for La2-HAp, 0.988(2)/0.009(3) for La3-HAp, 0.955(3) / 0.045(3) for La4-HAp. For La1-HAp, the precision of the La refined occupancy is poor due to the small doping percentage as well as the not optimal conditions for a better Rietveld refinement process. However, the profile reliability parameters (Rp, Rwp) point out the small percentage presence of La also in La1-HAp.'

- Probably  optical  fluorescence microscopy  or just fluorescence will  be needed to prove the doping not Raman /FTIR  - the  508 band is  very frail ( compared to   1000 and 630 cm-1.

We uploaded EDS qualitative spectra of the four samples in the Supporting Informations.

About the 508 cm-1 RE-O band, we have already noticed in a previous work of us on Eu-doped and Gd-doped hydroxyapatite that it is quite weak (Crystals 2020, 10, 806). Honestly, without knowing the position from literature, we would not have found it, but it is there.

- The weak broadening of Raman peaks for  RE presence ?  

The sentence reported in the Conclusions describes qualitatively the modest broadening in the four Raman spectra. It is not an indicator (presence) of rare earth. It is only describing the weak change of the spectra in the four samples, different among themselves for the Lanthanum doping.

- I am not sure that RE2O (oxide) will be structural incorporated in Hap. In the synthesis  procedure the  agate mortar for 1 hour; manually or by ball milling ?

The RE oxides are incorporated in HAp matrix, as qualitatively showed by EDS spectra (in the SI), and by the (frail but present) RE-O infrared band at 508 cm-1.

The synthesis procedure was a conventional solid state reaction by manual grinding for 1h  and calculated at T = 1300 C for 7h, as previously reported by our group working on Nd-HAp, Gd-HAp and Eu-HAp (Paterlini et al., Crystals 2020),

- homogenized powders were placed in alumina crucibles. Were they  pressed/pelleted? What were the amounts  used e.g. 1 g – 5 g 10 g total ?

All samples were placed in alumina crucibles and placed in the same furnace for the same time, in order to respect the same experimental conditions for all prepared samples. The amount used was 1g for each sample. Samples were not pressed and not pelleted. They are powders, always in agreement with previous work of us (Paterlini et al., Crystals 2020).

- XRF/XPS or other methods  providing  0.002 accuracy or better  is  a  must. Fluorescence is  also suited.

As stated before, we added additional SEM-EDS analyses in order to satisfy Reviewer’s comments. Qualitative EDS spectra are now placed in SI (Figures S6-S9).

Reviewer 4 Report

The manuscript by Rocco Cancelliere and co-authors reports on electrochemical and structural characterization of lanthanide doped hydroxyapatite. The authors prepared a series of rare-earth doped Ca10-xREx(PO4)6(OH)2 (RE = La, Nd, Sm, Eu, Dy, Tm; x = 0.01, 0.02, 0,10, 0.20) (HAp) powder samples. Electrochemical characterizations involved electrochemical impedance spectroscopy, cyclic voltammetry, and square wave voltammetry. The claimed outstanding sensitivity, repeatability, reproducibility, and reusability from the electrochemical characterizations are not based on solid evidence in my opinion.

The structural characterizations applied techniques including optical microscopy, X-ray diffraction on powder samples, and Infrared and Raman spectroscopies. The structural characterizations are not convincing in my opinion. The optical microscopy images didn’t show meaningful information on the samples, and the dots on the surface look like irregular dust contamination on the sample, electron microscopy with elemental analysis in this case seems more informative. The X-ray diffraction analysis shown in Figure 5 should be ascribed properly. the full comparison in one figure between observed PXRD and reported PXRD from JSPDS should be presented. The cell volumes are not changing with the lanthanum doping contents in my opinion, considering the experimental error. The structural discussion on the La3+ ions replacing Ca2 position with 2+ charge is not informative and most probably not reliable.

Based on the above concerns, I do not support the publication in its present form. However, after the authors clarify the concerns, I would like to re-evaluate the manuscript.

none

Author Response

- The claimed outstanding sensitivity, repeatability, reproducibility, and reusability from the electrochemical characterizations are not based on solid evidence in my opinion.

Thank you for your insight, but we disagree with this viewpoint. Table 2 presents the electrochemical characterization together with the signal's relative standard deviation and its percentage rise. Moreover, we provide more details on the findings of repeatability, reproducibility, and reusability, and sensitivity are depicted in Figures S1, S2, S3, and S4, respectively, in the Supporting Informations. Using the anodic peak current of each voltammogram, the analytical robustness of hydroxyapatite-modified SPE was estimated.

- The structural characterizations are not convincing in my opinion:

The following sentence was added to clarify results on structural characterizations:

'The presence of a high background signal at small 2q angles in all the La-HAP samples and of an additional crystalline phase even though with a small percentage prevent the refinement process from providing high-precision results. This does not lower the quality of the structural characterization presented in this paper.'

-The optical microscopy images didn’t show meaningful information on the samples, and the dots on the surface look like irregular dust contamination on the sample, electron microscopy with elemental analysis in this case seems more informative.

We added micrographs from SEM investigation in the text. We moved figures from optical microscopy in the Supporting Informations (Figure S5). About elemental analysis, we reported qualitative EDS spectra always in the SI (Figures S6-S9).

- The X-ray diffraction analysis shown in Figure 5 should be ascribed properly. the full comparison in one figure between observed PXRD and reported PXRD from JSPDS should be presented.

We added a new Figure (n. 6), with La3-HAp spectrum and with lines pertinent to the mentioned JCPDS entries (HAp, TCP, CaCO3).

- The cell volumes are not changing with the lanthanum doping contents in my opinion, considering the experimental error.

The following sentence was added:

'However, considering the standard deviation values, the small increase of the La-doping from La1-HAp to La2-HAp does not evidently change their volumes and unit cell parameters.'

- The structural discussion on the La3+ ions replacing Ca2 position with 2+ charge is not informative and most probably not reliable.

We thank the Reviewer for this criticism, but we introduced this point according to experimental results obtained by Fleet et al. 2000, Graeve et al. 2010, and by two works from our group (Baldassarre et al., 2020; Paterlini et al., 2020]. They all report RE cations in Ca2 site. We are describing La distribution in powder HAp samples (Fleet et al. worked on single crystals). About the discussion on bond valence parameters, we partially agree with the Reviewer. Bond valence parameters provide an indication, we can use the adjective ‘qualitative’ before bond valence investigation, but please note that such investigation is widely carried out in XRD characterization works.

Round 2

Author Response

For the synthesis for 1 g of synthesis material - how much RE-oxide has been weighted to obtain the 0.01 (unknown units)?

Answer

In order to obtain 1 g of Ca9.99La0.01(PO4)6(OH)2) sample (0.01 express atoms within crystal-chemical formula), we weighted 0.0016 g of La2O3, 0.399 g of CaCO3, and 0.81 g of CaHPO4, carefully measured with analytical balance.

If this is a “solid-state synthesis” what is the “0.05 M phosphate buffer saline (PBS), 0.1 M KCl, pH = 7.4,..” used for and how much? Besides why the EDS does not detect K?

Answer

We apologize with the Reviewer, this sentence is pertinent to the electrochemical measurements and not to the solid-state synthesis. It was a mistake for us to report this sentence in the synthesis paragraph. Now we deleted it. Sorry.

The provided EDS data regarding the presence of RE in the samples is not conclusive. The EDS are given as qualitative whereas the method is quantitative. Moreover, the EDS does not disclose/detect the presence of La in the samples (peaks for La are missing fig S6- S9). Either the EDS method is not the appropriate one or no La is present making the whole discussion unnecessary.

Answer

As regards the certain presence of lanthanum, we can say that the analyzed granules invariably show the presence of the rare earth, even if to different extents from sample to sample. It must be said that the quantity of metallization necessary for imaging is not suitable for quantitative analysis and therefore we do not allow ourselves to give numbers.

What is the EDS detector linked to the Zeiss EVO MA 10?

Answer

It is an EDS Bruker Quantax 200, LN2 free. We reported it in the Supporting Info.

The authors do not understand the basics of XRD Rietveld method especially the limits of the method and to what stage the output should be interpreted. The presence of additional phases in presented diffractogram makes the refinement to precision below 0.2 ( whatever the nominal unit is) unreliable.

Answer

We have modified two sentences in the text in order to take into account what suggested from the Reviewer.

Line 323: ‘The presence of a high background signal at small 2θ angles in all the La-HAP samples and of an additional crystalline phase even though with a small percentage prevent the refinement process from providing high-precision results. This does not lower the quality of the structural characterization presented in this paper’.

Changed with: ‘The presence of a high background signal at small 2θ angles in all the La-HAP samples and of an additional crystalline phase even though with a small percentage prevent the refinement process from providing high-precision results and lower the reliability of the structural characterization presented in this paper’.

Line 410: ‘For La1-HAp, the precision of the La refined occupancy is poor due to the small doping percentage as well as the not optimal conditions for a better Rietveld refinement process. However, the profile reliability parameters (Rp, Rwp) point out the small percentage presence of La also in La1-HAp.’

Changed with: ‘The precision of the variables refined by the Rietveld method, in particular of the La occupancy for La1-HAp, is poor due to the small doping percentage, as well as the presence of a high background signal at small 2θ angles and an additional crystalline phase’.

In the response the authors have replied that CIFs were provided in the SI, but no such information is present.

Answers:

We had uploaded single CIFs, as ‘files’ and not within SI as text. But now we realized that there are no uploaded cif files: we are sorry, we think it was a problem with the electronic procedure. We had the same problem with the graphical abstract. Now we placed the four CIF files within SI as text.

Tried to reach the structures (CIFs) through the database, but those are not available.

Answer:

The CIF files are not yet available in joint CSD-CCDC database until the paper is published.

Besides only 2 deposition numbers are provided for 4 structures?

Answer:

We apologize with Reviewers for this misprinting, the deposition numbers are four and they are exactly: CSD2250357 (La1-HAp), CSD2250358 (La2-HAp), CSD2250362 (La3-HAp) and CSD2250363 (La4-HAp) (we revised the text accordingly)

Besides why in table 3 the La is not present

Answers

The topic of Table 3 is an introductive discussion about crystallographic sites present in all hydroxyapatite, not only in the present samples. Later, we talk about the presence of dopant lanthanum (lines 404-417 of the revised manuscript). Anyway, the discussion of this table is fixed according to the structural arrangement of hexagonal P63/m hydroxyapatite, with atoms occupying distinct crystallographic sites, i.e. Ca1 in 4f special site, Ca2, P1, O1 and O2 on 6h special site, O3 on 12i general position, and O4OH on 4e special site (Table 3). We better specified those aspects in the text.

what are those x y z coordinates (= 1 or 0 or …. 1=0) in the in the table 3?

Answer

x y z are the atomic coordinates taken from CIF file. They usually range among 0 and 1 within asymmetric unit of unit cell, and when the atom is placed on a site with symmetry operators, some of these coordinates can be fixed numbers such as 0.250, 0.333 etc. They can also exceed 1, by adding or subtracting integers numbers (1, 2, ecc) due to symmetry of unit cell.

Why the P1, Ca2 and O1 have the same coordinates ?

Answer:

P1, Ca1, and O1 have only the same z coordinate (0.250), but x and y can both vary.

The Electrochemical characterization e.g. cyclic voltammetry for the so-called La-HAP should be compared with undoped HAP to prove that there is a difference.

Answer

We appreciate the Reviewers comment. Nonetheless, we disagree. The electrochemical characterization was performed on unmodified HAp and HAp-modified electrodes. Observing the results presented in Table 2, it is possible to determine the value of Rct, which is the difference between the Rct of undoped and doped HAp.

The conclusions about weak broadening of Raman peaks and the peak at ~508 cm-1 in the FTIR should be removed.

Answer

We decided to remove the short comment on Raman peak, but to keep that on FTIR peak at 508 cm-1 according to qualitative finding of La presence from EDS analysis, and to the referred bibliography (Ref. 66 = doi:10.1021/cm001117p): we only changed the pertinent sentence in ‘The structural characterization, based on X-ray diffraction, FTIR and Raman spectroscopies, showed increasing unit cell dimensions according to PXRD data, and a band mode in FTIR spectra at 505-508 cm-1 reasonably assigned to RE-O bond’.

As a general remark, even if we disagree with Reviewer 3 for his/her opinion on our qualitative EDS analysis, we thank him/her to pushing us to improve the manuscript.

Regards, F. Capitelli, R. Cancelliere, and co-authors.

Reviewer 4 Report

The revision resolved my concerns, I support the publication.

Author Response

The revision resolved my concerns, I support the publication.

We thank Reviewer 4 for his/her evaluation of our revised manuscript.

F. Capitelli, R. Cancelliere and co-authors

Round 3

Reviewer 3 Report

I am  still pusled by the discrimination between 0.01 and 0.02 La content using PXRD.

For me, even samples preparation - e.g. two different PXRD preparations of the same material -  will produce an error that is several times 0.01.

The employed EDX/EDS does not detect La in the  samples, ... 

Thats  all. 

Author Response

Regarding the first comment:

For me, even samples preparation - e.g. two different PXRD preparations of the same material -  will produce an error that is several times 0.01.

We are aware that in some samples the La doping percentage is low. However, even low lanthanum doping percentages produce a statistically significant variation of the electrochemical detection signal. This result is of outstanding relevance for the development of sensitive La-Hap based sensors, which motivated our work.

However, we have further modified the following sentence from:

‘The precision of the variables refined by the Rietveld method,

in particular of the La occupancy for La1-HAp, is poor due to the

small doping percentage, as well as the presence of a high

background signal at small 2θ angles and an additional crystalline

phase’.

to

The precision of the variables refined by the Rietveld method, in 

particular of the La occupancy for La1-HAp, is poor due to the small

doping percentage, as well as the presence of a high background

signal at small 2θ angles and an additional crystalline phase,

making questionable the presence of La in the lowest doped sample.'

Regarding the second comment:

The employed EDX/EDS does not detect La in the  samples, ... Thats  all.

We would like to specify that:

The heterogeneous morphology of the granes makes difficult the readability of EDS qualitative results, partially interfering with the X-ray fluorescence emitted by the samples.

Nevertheless, EDS is able to detect Rare Earth content up to 0.1 wt % in crystalline matrices.